# Comparison of Surgical Outcomes between Single-Use and Reusable Flexible Ureteroscopes for Renal Stone Management: A Systematic Review and Meta-Analysis

**DOI:** 10.3390/medicina58101388

**Published:** 2022-10-03

**Authors:** Dae Young Jun, Kang Su Cho, Jae Yong Jeong, Young Joon Moon, Dong Hyuk Kang, Hae Do Jung, Joo Yong Lee

**Affiliations:** 1Department of Urology, Severance Hospital, Urological Science Institute, Yonsei University College of Medicine, Seoul 03722, Korea; 2Department of Urology, Prostate Cancer Center, Gangnam Severance Hospital, Urological Science Institute, Yonsei University College of Medicine, Seoul 06273, Korea; 3Department of Urology, Inha University College of Medicine, Incheon 22212, Korea; 4Department of Urology, Inje University Ilsan Paik Hospital, Inje University College of Medicine, Goyang 10380, Korea; 5Center of Evidence Based Medicine, Institute of Convergence Science, Yonsei University, Seoul 03722, Korea

**Keywords:** disposable equipment, kidney calculi, meta-analysis, systematic review, ureteroscopes

## Abstract

*Background and Objectives*: Disposable flexible ureteroscopes have been widely used because of their cost-effectiveness and higher sterility potential compared with reusable flexible ureteroscopes. This study aimed to compare the surgical outcomes and complication rates in patients who undergo reusable or disposable flexible ureteroscopic stone surgeries (fURS) for urinary stone disease. *Materials and Methods*: A systematic review and meta-analysis were conducted under the Preferred Reporting Items for Systematic Reviews and Meta-Analyses (PRISMA) guideline. This systematic review was registered with PROSPERO (CRD42022331291). Clinical trials comparing reusable and disposable fURS for stone disease were found from PubMed, EMBASE, Cochrane Library, and the Web of Science up to March 2022. Participants were patients with upper urinary tract stones; the interventions were reusable or disposable fURS. Outcomes, including stone-free rate, operation time, length of hospital stay, and complication rate, were compared for analysis. *Results*: Overall, 111 studies were identified, but after removing duplicate studies, 75 studies remained. Thirty-two of these studies were excluded. Of the 43 screened studies, 11 met the eligibility criteria. There was no difference in the stone-free rate (SFR) between disposable and reusable fURS (*p* = 0.14; OR = 1.36; 95% CI, 0.9 to 2.04). For operation time, no difference was identified between reusable and disposable fURS groups (*p* = 0.12; MD = −5.31; 95% CI, −12.08 to 1.46). For hospital stay, there was also no difference between the two groups (*p* = 0.61; MD = −0.03; 95% CI, −0.17 to 0.10). There was no significant difference in complication rate between the two groups (*p* = 0.85; OR = 0.95; 95% CI, 0.56 to 1.61). *Conclusions*: There were no differences in the SFR, operation time, length of hospital stay, and complication rate between reusable and disposable fURS. Disposable fURS may be a comparable alternative to reusable fURS.

## 1. Introduction

Since 1990, cases of urolithiasis have been increasing worldwide [1]. The overall kidney stone prevalence in the United States increased from 3.2% to 10.1% from 1980 to 2016 [2]. The hospital incidence of ureter and renal stones increased by 63% in the United Kingdom from 2000 to 2010 [3]. The prevalence of urolithiasis has increased significantly in most Asian countries over the past few decades (China, from 4% to 6.4%; Japan, from 4.3% to 9.0%; South Korea, from 3.5% to 11.5%) [4]. Therefore, it is important to determine the most effective and safest treatment modality for the health and quality of life of urolithiasis patients [5].

In the European Association of Urology Guidelines on Urolithiasis, flexible ureteroscopy is the treatment of choice for renal stones smaller than 20 mm and possible second-line therapy for renal stones larger than 20 mm [6,7]. The flexible ureteroscopy has a wide range of applicable stone sizes and can be applied to renal and ureter stones [8].

The first modern flexible ureteroscopes with a two-way deflection and working channel were introduced in 1994 [9]. In addition, the introduction of the holmium: yttrium-aluminum-garnet laser system, the improvement of image quality and durability, and the extension of deflection angles have made flexible ureteroscopy widely available and accessible to the entire renal calyces [10]. In 2015, the first disposable digital flexible ureteroscope was introduced [11]. Since then, single-use flexible ureteroscopes have been used for diagnostic purposes, biopsy, or stone removal, and many studies have been conducted on clinical outcomes, including the stone-free rate (SFR) and maintenance cost of reusable versus single-use flexible ureteroscopes [12,13]. However, there is limited concrete clinical evidence on the relative advantages and disadvantages of reusable versus disposable flexible ureteroscopes [14]. An updated meta-analysis was performed to compare surgical outcomes in patients who underwent reusable or single-use flexible ureteroscopic stone surgeries (fURS) for the removal of upper urinary tract stones. This study will affirm the reliability of disposable fURS, with the addition of updated evidence through meta-analysis of recent studies.

## 2. Materials and Methods

### 2.1. Inclusion Criteria

This systematic review and meta-analysis followed the participants, interventions, comparators, outcomes, study design (PICOS) approach, and PRISMA guidelines (Appendix A) [15]. The inclusion criteria of articles for this study were as follows: (a) patients with upper ureter and kidney stones, (b) intervention and comparators including fURS using reusable and disposable flexible ureteroscope, and (c) measures of clinical outcomes including the SFR, operation time, length of hospital stay (LOS), or complication rate. In the study design, our analysis included both randomized controlled trials (RCTs) and non-RCTs. Published studies without available full text were excluded. This systematic review was exempt from review by the ethics committee or institutional review board because systematic reviews and meta-analyses do not require ethical approval.

### 2.2. Search Strategy

A systematic review was performed to identify relevant articles that compared treatment for renal stones using the four English language databases PubMed, EMBASE, the Cochrane Central Register of Controlled Trials (Central), and Web of Science up to March 2022. Search strategies were established to include Medical Subject Headings keywords such as “ureteroscopy”, “flexible ureteroscope”, “single-use”, “disposable”, “reusable”, “upper urinary calculi”, “urolithiasis”, “kidney stone”, “ureteral calculi”, and combinations of these search terms.

### 2.3. Study Selection and Extraction

Our researchers screened titles and abstracts and independently identified by the search strategy to exclude irrelevant studies. They also assessed the full text of the articles to search for potentially relevant articles. The most relevant articles were extracted from each study, and information such as author, the year of publication, the country, study design, patient characteristics, and treatments as well as outcome variables such as “SFR”, “operation time”, “length of hospital stay”, and “complication rate” was recorded.

### 2.4. Quality Assessment

Cochrane risk of bias (ROB) tool was used for RCTs, and the methodological index for nonrandomized studies (MINORS) was used for nonrandomized studies. The quality of evidence was graded using the Scottish Intercollegiate Guidelines Network (SIGN), which is composed of various types of research articles, including systematic reviews and meta-analyses, RCTs, cohort studies, case–control studies, diagnostic studies, and economic studies. A quality assessment was carried out by our researchers independently (D.Y.J. and Y.J.M.). All disagreements regarding the quality assessment results were cleared up after discussion with a third reviewer (J.Y.L.).

### 2.5. Statistical Analysis

The odds ratios (ORs) and 95% confidence intervals (CIs) were calculated and reported for dichotomous variables. The mean differences (MD) and 95% CIs were calculated for the continuous variables. The chi-squared test, with *p*-values less than 0.05, was used to evaluate statistical heterogeneity, and the *I*^2^ statistic was used to quantify heterogeneity [16]. If the reported *I*^2^ statistic was less than 50%, we applied the fixed-effects model; otherwise, the random-effects model was used. The Higgins *I*^2^ statistic was calculated as follows:$$I^{2}=\frac{Q-df}{Q}\times 100\%$$
where *Q* is the Cochrane heterogeneity statistic, and df is the degrees of freedom. A meta-analysis was conducted with R software, version 4.1.3 (R Foundation for Statistical Computing, Vienna, Austria; http://www.r-project.org) and its meta package and Review Manager, version 5.4.1 (RevMan, Copenhagen, Denmark: The Nordic Cochrane Center, The Cochrane Collaboration, Oxford, UK, 2020). In the analysis of SFR, operation time, length of hospital stay, forest plots were used to visually display results, funnel plots were used to assess reporting bias, and L’Abbe plot or radial plots were used to assess heterogeneity. In addition, sensitivity tests using outcome reporting bias were conducted. A subgroup analysis was performed between the reusable and disposable fURS groups according to the type of reusable flexible ureteroscopes (fiber-optic or digital). This systematic review was registered with PROSPERO (CRD42022331291).

## 3. Results

### 3.1. Eligible Studies

After the review of all the original articles, 11 studies related to this meta-analysis were identified (Figure 1) [12,17,18,19,20,21,22,23,24,25,26]. The characteristics of the 11 included studies are shown in Table 1 [12,17,18,19,20,21,22,23,24,25,26]. All these comparative studies included patients who underwent fURS for upper ureter and renal stones with reusable or disposable flexible ureterorenoscope. The publication date of the included studies was from August 2015 to March 2022. Four of the eleven studies were performed in China [17,20,23,26]; two, in Germany [18,21]; one, in Australia [12]; one, in Chile [24]; one, in Greece [22]; one, in Turkey [19]; and one in the US [25].

### 3.2. Quality Assessment and Publication Bias

The quality assessment results according to the SIGN are shown in Table 1. Funnel plots of this meta-analysis are shown in Appendix A. Most of the included studies were located in the funnel with respect to the SFR and the LOS, except for operation time. The ROB for three RCTs is shown in Appendix A. The MINORS scores for all the nonrandomized studies are displayed in Appendix A. All studies were thought of as reasonable with respect to study quality and publication bias.

### 3.3. Heterogeneity Assessment

There was little heterogeneity in the heterogeneity test for the SFR between the digital reusable fURS and disposable fURS groups, so the fixed-effects model was fulfilled (Figure 2B Forest plot). However, high heterogeneity was identified in the analysis of the SFR in the whole (Both fiber-optic and digital) reusable fURS or fiber-optic reusable fURS and disposable fURS groups (Figure 2A,C Forest plot), so random-effects models were applied. In the meta-analysis of SFR in each study design, heterogeneity was high and random-effects model was applied (Figure 3).

In the analysis of the operation time of digital reusable fURS and disposable fURS, little heterogeneity was identified, so the fixed-effects model was applied (Figure 4B Forest plot). For comparison of operation time between whole reusable fURS or fiber-optic reusable fURS and disposable fURS, the random-effects model was used because of the apparent heterogeneity (Figure 4A,C Forest plot). The fixed-effects model was used in the analysis for LOS because there was little heterogeneity (Figure 5 Forest plot).

A L’Abbe plot is shown in Appendix A, and three of the eleven studies showed a little heterogeneity. Three radial plots are shown in Appendix A. A radial plot of the SFR shows that two out of 10 studies are heterogeneous (Appendix A). A radial plot of the operation time shows that there is some heterogeneity in four out of nine studies (Appendix A). A radial plot of LOS from seven studies shows no heterogeneity (Appendix A).

A sensitivity analysis for outcome reporting bias (ORB) to examine the degree of heterogeneity was conducted (Appendix A). The outcomes for the SFR, operation time, and LOS were not affected until up to three studies were excluded.

### 3.4. Stone-Free Rate

Ten studies were included for the SFR [17,18,19,20,21,22,23,24,25,26]: three studies were RCT [17,23,26], and seven studies were non-RCT [18,19,20,21,22,24,25]. There was no difference in the SFR between whole reusable fURS and disposable fURS (*p* = 0.14; OR = 1.36; 95% CI, 0.9 to 2.04; Figure 2A). A subgroup analysis that included six studies between the fiber-optic reusable fURS and disposable fURS groups showed the same results (*p* = 0.59; OR = 1.21; 95% CI, 0.61 to 2.42; Figure 2C). However, when comparing four studies, the SFR of the disposable fURS group was higher than that of the digital reusable fURS group (*p* = 0.03; OR = 1.64; 95% CI, 1.05 to 2.56; Figure 2B). In a subgroup analysis of SFR according to study design, there was no difference of SFR between each study design of RCT, prospective study, and retrospective study (*p* = 0.14; OR = 1.36; 95% CI, 0.90 to 2.04; Figure 3). Additionally, there was no subgroup difference (*p* = 0.90).

### 3.5. Operative Time

Nine studies were included for operation time [17,19,20,21,22,23,24,25,26]: three studies were RCT [17,23,26], and six studies were non-RCT [19,20,21,22,24,25]. Between the whole reusable fURS and disposable fURS groups, there was no difference in operation time (*p* = 0.12; MD = −5.31; 95% CI, −12.08 to 1.46; Figure 4A). The same results were shown in a subgroup analysis between the digital reusable fURS and disposable fURS groups (*p* = 0.70; MD = 0.74; 95% CI, −2.95 to 4.42; Figure 4B) and between fiber-optic reusable fURS and disposable fURS groups (*p* = 0.07; MD = −10.14; 95% CI, −21.27 to 0.99; Figure 4C).

### 3.6. Length of Hospital Stay

Seven studies were included for LOS [17,19,20,22,23,25,26]: three studies were RCT [17,23,26], and four studies were non-RCT [19,20,22,25]. Between the whole reusable fURS and disposable fURS groups, there was no difference in LOS (*p* = 0.61; MD = −0.03; 95% CI, −0.17 to 0.10; Figure 5A). The same results were shown in a subgroup analysis between the digital reusable fURS and disposable fURS groups (*p* = 0.23; MD = −0.15; 95% CI, −0.40 to 0.10; Figure 5B) and between the fiber-optic reusable fURS and disposable fURS groups (*p* = 0.92; MD = 0.01; 95% CI, −0.14 to 0.16; Figure 5C).

### 3.7. Complication Rate

Four out of eleven studies classified complications by Clavien–Dindo classification (Table 1). In a comparison of complications between Clavien–Dindo classification I~II and III~IV, the complication rate did not show any significant difference between the two groups (*p* = 0.85; OR = 0.95; 95% CI, 0.56 to 1.61) (Appendix A).

## 4. Discussion

A deflectable ureteroscope was described in 1964, but unlike today’s ureteroscopes, it was passively deflected [27]. In 1971, the flexible ureteroscope was inserted through the urethra up to the ureter and renal pelvis, and the tip of the ureteroscope was capable of bidirectional active deflection up to 30 degrees [28]. In 1987, the first disposable flexible ureteroscope was introduced; it had a modular design, and the shaft tip was passively deflected [29].

There are several problems with reusable flexible ureteroscopes. First, high acquisition and repair costs can be a burden to maintaining a flexible ureteroscope [30]. Second, flexible ureteroscopes are repeatedly broken and require repair; a repaired flexible ureteroscope breaks four times faster than a new one [31]. Several factors influencing the fragility of flexible ureteroscopes cannot be altered, such as the amount of actuation time spent at the lower pole and repeated passage of the instrument through the working channel [32,33]. However, other issues, including laser activation in the working channel and increased use of force, can be addressed through education and training [34]. Third, one report found cultured pathogens in reusable flexible ureteroscopes even after sterilization [35]. It has not been proven whether culture positivity in sterilized reusable flexible ureteroscopes increases the risk of urinary tract infection in patients who will be treated with these ureteroscopes, but the probability of infectious complications might not be completely eliminated.

The single-use flexible ureteroscope is an alternative to the reusable flexible ureteroscope. If a single-use flexible ureteroscope is used, recurrent damage to instruments is avoided [36]. Thus, labor costs for maintaining reusable flexible ureteroscope can be reduced. Cost-effectiveness can be one of important factor to decide treatment modality in some countries [37]. Some studies have calculated the break-even point between disposable flexible ureteroscopes and reusable flexible ureteroscopes. [38,39]. In these studies, disposable flexible ureteroscopes are more cost-effective than reusable flexible ureteroscopes when there are less than 22 to 99 cases of ureteroscopic surgeries, favoring the use of disposable flexible ureteroscopes in smaller centers. Large stones in the lower pole of the kidney and a steep infundibulopelvic angle are risk factors for damage to flexible ureteroscopes. A possibility of injuries of flexible ureteroscope exists during lasering in those kidney stones. In these cases, single-use flexible ureteroscopes can be a cost-effective alternative to reusable flexible ureteroscopes [40]. There was a report of insufficient sterilization after the use of reusable flexible ureteroscopes [41]. The use of a disposable flexible ureteroscope does not include the risk of patient-to-patient infection because there is no cleaning and disinfection process.

In this study, the SFR, operation time, and LOS were compared between single-use and reusable flexible ureteroscopes. There was no statistically significant difference. Overall, the findings of this study indicate that the single-use flexible ureteroscope is a comparable alternative to the reusable flexible ureteroscope.

In the subgroup analysis, there was no statistically significant difference in the SFR between the reusable fiber-optic ureteroscope group and the single-use ureteroscope group. However, the SFR was higher in the single-use ureteroscope group than in the reusable digital ureteroscope group, and the result was statistically significant. This finding may be due to the different deflection angles in the initial and more recent digital flexible ureteroscopes. Although most flexible ureteroscopes in this study had a bidirectional deflection angle of 270–275 degrees, the reusable fiber-optic flexible ureteroscope URF-P5 (Olympus Corp., Tokyo, Japan) and the reusable digital flexible ureteroscope URF-V (Olympus Corp.) have a 180-degree upward deflection angle. The proportion of patients treated with reusable flexible ureteroscopy with a relatively low angle of deflection can influence the SFR. Four studies were analyzed in the comparative pooled analysis of reusable digital ureteroscopy versus single-use flexible ureteroscopy. In three of these studies, 206 (91.6%) patients were treated with URF-V from a total of 225 in the reusable digital flexible ureteroscope group (Figure 4B). Six studies were analyzed in a comparative pooled analysis of reusable fiber-optic flexible ureteroscopy versus single-use flexible ureteroscopy and in a single study, URF-P5 was used. In this subgroup analysis, 180 of 390 patients (46.2%) who underwent reusable fiber-optic flexible ureteroscopy were treated with the URF-P5. In 2015, Ding et al. used the single-use flexible ureteroscope Polyscope (PolyDiagnost, Hallbergmoos, Germany), which can be deflected unidirectionally 265 degrees [17]. In their study, Polyscope was compared with URF-P5. In the comparative analysis, the inferiority of the deflection function of Polyscope and URF-P5 might have been able to offset the negative effect on the SFR. Another study suggested that the fiber-optic flexible ureteroscope’s accessibility to the sharp-angled calyx is superior to that of a digital flexible ureteroscope because of the narrow diameter of the fiber-optic scope, and these characteristics may have influenced the results of our analysis [42]. Several factors affecting the SFR, such as stone size criteria, stone location, imaging modality, and timing of follow-up imaging, are not homogeneous. Thus, those heterogeneities can be some limitations of the analyses [43,44,45,46].

Seven of the eleven studies reported specified complication profiles (Appendix A). Based on the modified Clavien classification system (MCCS), each complication was classified as MCCS grade I~II and III~IV. Nine categories of complications were compared by the chi-squared and Fisher’s exact test. There was no statistically significant difference in complication rates between the disposable and reusable fURS groups.

Infection may be an important factor when selecting reusable or single-use flexible ureteroscope because the use of sterilized reusable ureteroscopes is associated with a small risk of infection [35]. There are several components of complications related to infection. For example, fever, urinary tract infection, and sepsis can be infectious complications [47]. Sepsis, which can be life-threatening [48], was described in three of the eleven studies. For these three studies, the number of patients who developed sepsis in the single-use versus reusable ureteroscope groups was 9 (5%) versus 6 (3.3%) (*p* = 0.599), 0 versus 4 (11%) (*p* = 0.049), and 5 (4.2%) versus 5 (4.2%) (*p* = 1.000) [17,20,22]. Uniquely, Mourmouris et al. reported no infection in their disposable flexible ureteroscope group and showed a significant difference in infection rate [22]. Their results might be weak evidence of the advantage of disposable flexible ureteroscope against perioperative infection. When considering the possibility of ureteroscope contamination after sterilization reported previously, a single-use, flexible ureteroscope can be preferred. However, there have been controversies in previous studies [49]. More studies should be performed on infectious complications from the use of single-use versus reusable flexible ureteroscopes.

There was a previous study comparing disposable and reusable fURS using meta-analysis [50]. Although many studies on fURS have been conducted since that study, only a few studies were included in the current analysis, as the results of some studies were not relevant to our study [51]. The studies not including SFR, operation time, and length of hospital stay did not satisfy the inclusion criteria and were excluded. Among the studies not included, there were studies that dealt with important topics not covered in this study. First, the economic aspects of disposable and reusable fURS are important in that they influence the choice between disposable and reusable fURS. A previous study analyzed the cost per procedure of reusable flexible ureteroscope and concluded that reusable flexible ureteroscope is more cost-effective than disposable flexible ureteroscope [52]. Second, the importance of environmental aspects has been emphasized. A previous study examined the equivalent mass of carbon dioxide emitted each time a disposable or reusable flexible ureteroscope is used, and found that the environmental impact was comparable between disposable and reusable flexible ureteroscopes [53].

There are several limitations to this study. First, only 11 studies were included. Three of them were RCTs, seven were prospective studies, and one study was a retrospective study. Second, different flexible ureteroscopes were used in each study. Each flexible ureteroscope has different characteristics, such as deflection angle, weight of scope, mode of image transmission affecting the resolution of image, durability, diameter of scope, and diameter of working channel. Third, the length of time to image follow-up after stone surgery and modalities for imaging were different between the included studies. Therefore, the results of this study should be considered along with its limitations.

## 5. Conclusions

This meta-analysis found no differences in the SFR, complication, operation time, and hospital stay between reusable and disposable fURS. However, compared with digital reusable fURS, disposable fURS had a higher SFR. The type of digital reusable fURS in most of the included studies was URF-V, which has a limited deflection of fURS, and this limitation may cause differences in the SFR. Disposable fURS may be a comparable alternative to reusable fURS. However, more research is needed to draw definitive conclusions.

## Figures and Tables

**Figure 1 medicina-58-01388-f001:**
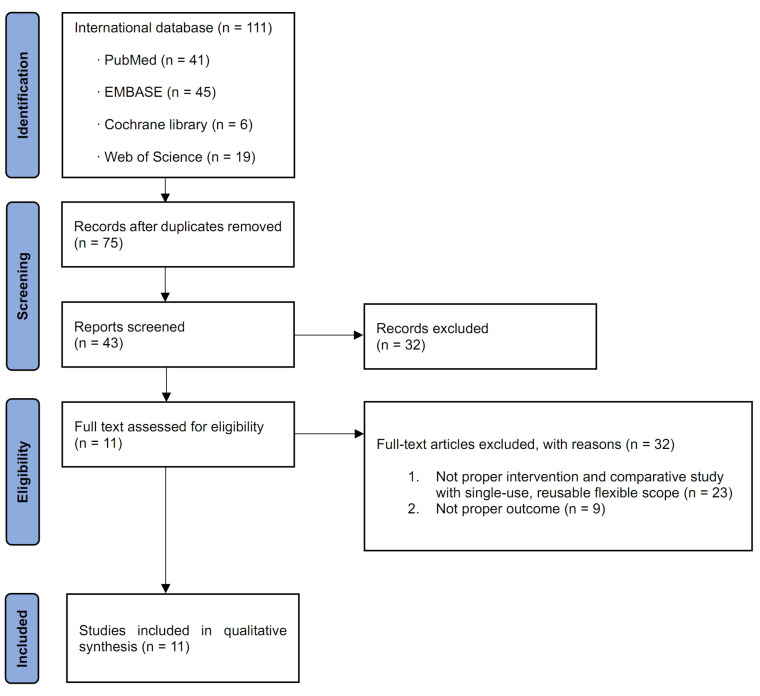
Study selection flow chart.

**Figure 2 medicina-58-01388-f002:**
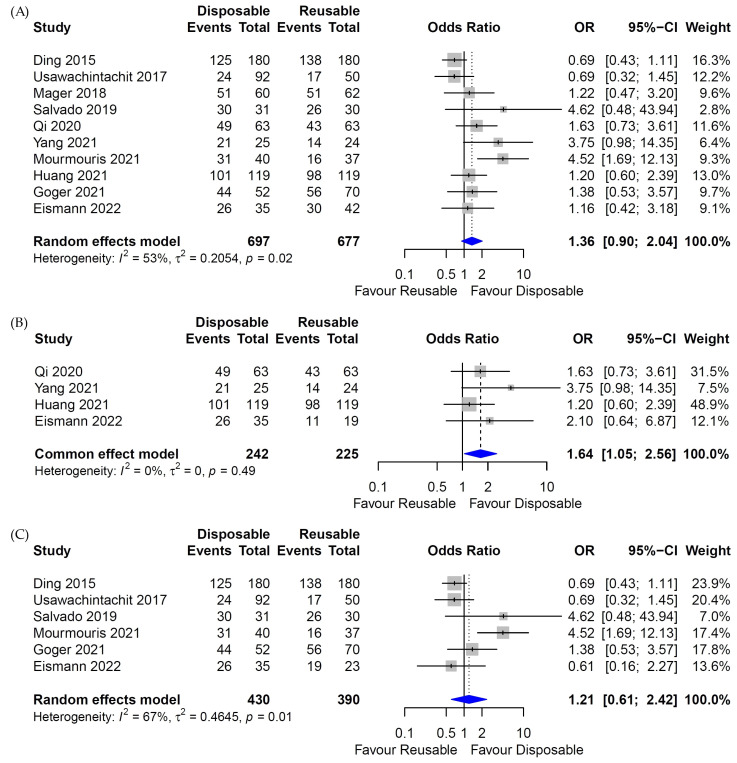
Forest plot for stone-free rate (SFR) between disposable and reusable flexible ureteroscopic stone surgeries (fURS). (**A**) SFR of disposable and whole reusable fURS. (**B**) SFR of disposable and digital reusable fURS. (**C**) SFR of disposable and fiber-optic reusable fURS [17,18,19,20,21,22,23,24,25,26].

**Figure 3 medicina-58-01388-f003:**
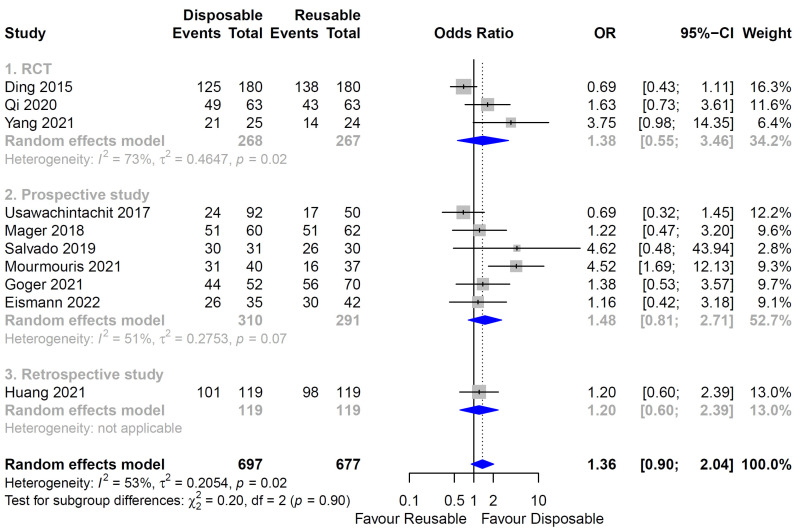
Forest plot for stone-free rate (SFR) of disposable and reusable flexible ureteroscopic stone surgeries (fURS) in each study design. (1) SFR of disposable and reusable fURS in randomized clinical trials (RCTs). (2) SFR of disposable and reusable fURS in prospective studies. (3) SFR of disposable and reusable fURS in retrospective study. Study design is written in gray above each analysis (RCT, prospective study, retrospective study). The heterogeneity of each analysis and the applied model are written in gray below each analysis. The heterogeneity between subgroups and the applied model are written in black at the bottom [17,18,19,20,21,22,23,24,25,26].

**Figure 4 medicina-58-01388-f004:**
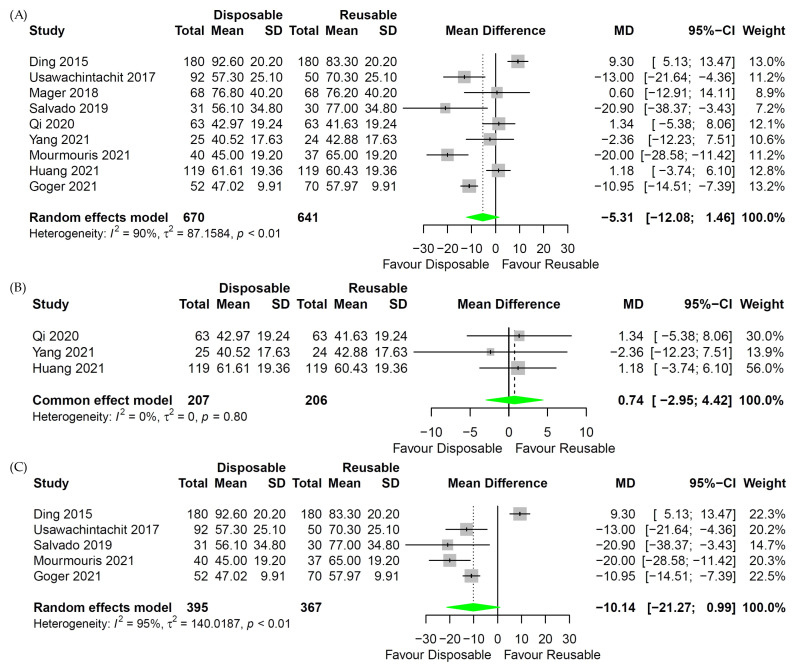
Forest plot for operation time between disposable and reusable flexible ureteroscopic stone surgeries (fURS). (**A**) Operation time of disposable and whole reusable fURS. (**B**) Operation time of disposable and digital reusable fURS. (**C**) Operation time of disposable and fiber-optic reusable fURS [17,19,20,21,22,23,24,25,26].

**Figure 5 medicina-58-01388-f005:**
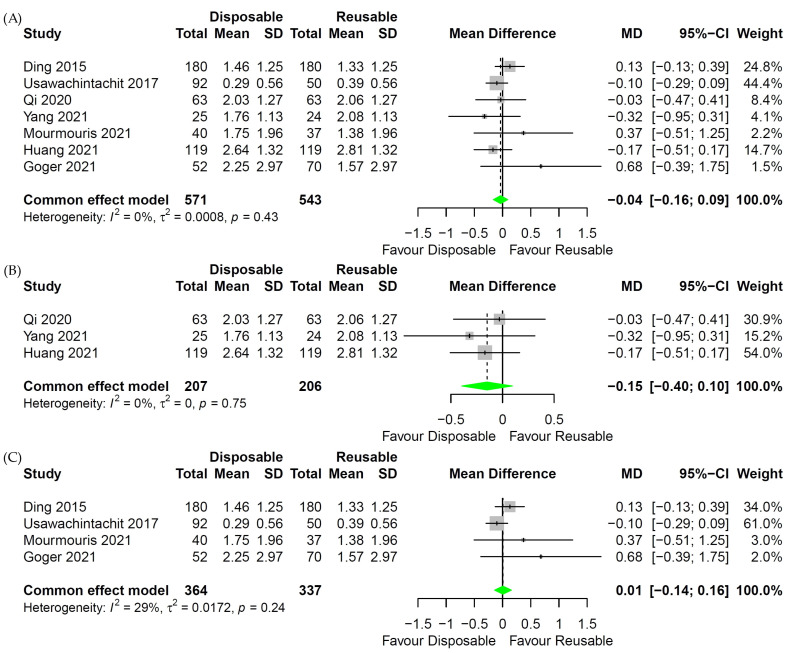
Forest plot for length of hospital stay (LOS) between disposable and reusable flexible ureteroscopic stone surgeries (fURS). (**A**) LOS of disposable and whole reusable fURS. (**B**) LOS of disposable and digital reusable fURS. (**C**) LOS of disposable and fiber-optic reusable fURS [17,19,20,22,23,25,26].

**Table 1 medicina-58-01388-t001:** Characteristics of the studies included in the analysis.

Author Year	Country	Study Design	Inclusion Criteria	Type of Scope		No. Patients	Age (Years)	Follow-Up	Definition of Stone-Free	Complications (No.)		Quality Assessment (SIGN)
										**I–II**	**III–IV**	
Ding et al., 2015 [17]	China	RCT	Renal stones < 3 cm	Disposable fURS	PolyScope	180	50.5 ± 12.8	1 day (4 weeks, if not acquire complete stone-free status)	≤4 mm	Not stated	Not stated	2+
				Reusable fiber-optic fURS	URF P-5	180	51.1 ± 13.7					
Usawachintachit et al., 2017 [25]	USA	Prospective	Not stated	Disposable fURS	LithoVue	92	55.8 ± 15.1	3 months	No fragments present (Insignificant residual fragment: <2 mm)	Not stated	Not stated	2+
				Reusable fiber-optic fURS	URF-P6	50	50.5 ± 12.6					
Mager et al., 2018 [21]	Germany	Prospective	Not stated	Disposable fURS	Lithovue	68	54 ± 17	Not stated	Not stated	9	3	2+
				Reusable fiber-optic/digital fURS	Flex-X2S, Flex-XC	68	59 ± 16			4	1	
Kam et al., 2019 [12]	Australia	Prospective	Patients undergoing retrograde fURS for the management of renal calculi, tumors, and ureteric strictures or ECIRS	Disposable fURS	Lithovue	55	53.5	Not stated	Not stated	8	0	2+
					PU3022A	31	54.1			9	0	
				Reusable digital fURS	URF-V2	64	53.3			12	0	
Salvado et al., 2019 [24]	Chile	Prospective	Single stone located in the inferior calyx	Disposable fURS	Uscope3022	31	50.4 ± 13.8	1 month	≤1 mm	Not stated	Not stated	1+
				Reusable fiber-optic fURS	Cobra	30	49.9 ± 16.5					
Qi et al., 2020 [23]	China	RCT	Maximum stone diameter 6~20 mm Renal stone or upper ureter stone	Disposable fURS	ZebraScope	63	51.84 ± 13.16	1 month	≤4 mm	Not stated	Not stated	2+
				Reusable digital fURS	URF-V	63	53.25 ± 12.11					
Yang et al., 2021 [26]	China	RCT	Maximum stone diameter 6~20 mm Lower pole stone	Disposable fURS	ZebraScope	25	52.72 ± 11.79	1 month	≤4 mm	Not stated	Not stated	2+
				Reusable digital fURS	URF-V	24	54.00 ± 12.69					
Mourmouris et al., 2021 [22]	Greece	Prospective	Radiopaque stone disease	Disposable fURS	Lithovue	40	55.73 ± 13.47	1 day	≤2 mm	Not stated	Not stated	1+
				Reusable fiber-optic fURS	Flex-X2	37	55 ± 11.2					
Huang et al., 2021 [20]	China	Retrospective	Upper urinary calculi	Disposable fURS	ZebraScope	119	49.4 ± 12.7	1 month	≤3 mm	13	1	1+
				Reusable digital fURS	URF-V	119	49.0 ± 12.0			9	3	
Göger et al., 2021 [19]	Turkey	Prospective	Lower pole stones	Disposable fURS	Uscope3022	52	52.4 ± 19.4	2~4 weeks	≤3 mm	9	0	1+
				Reusable fiber-optic fURS	Flex-X2S	70	48.73 ± 14.7			6	1	
Eismann et al., 2022 [18]	Germany	Prospective	Urolithiasis	Disposable fURS	PU3022A	35	Not stated	Not stated	≤1 mm	Not stated	Not stated	2+
				Reusable digital fURS	Flex-XC	19						
				Reusable fiber-optic fURS	Flex-X2S	23						

ECIRS; endoscopic-combined intrarenal surgery; fURS, flexible ureteroscopic stone surgeries; RCT, randomized controlled trial; SIGN; Scottish Intercollegiate Guidelines Network.

## Data Availability

The data presented in this study are available in the article.

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
