# Peer review of "Comparison of Surgical Outcomes between Single-Use and Reusable Flexible Ureteroscopes for Renal Stone Management: A Systematic Review and Meta-Analysis"

_medicina, 2022, doi:10.3390/medicina58101388_

Round 1
Reviewer 1 Report (Previous Reviewer 3)
The economic aspects are still discussed not broadly enough. Several aspects (e.g. break-even points reusable vs. disposable scopes) should be included as it is of great importance for users (cf. the literature e.g. Martin et al., J Urol 2017, Al-Balushi et al. Int UrolNephrol 2019).
Author Response
Reviewer 1
Comment: The economic aspects are still discussed not broadly enough. Several aspects (e.g. break-even points reusable vs. disposable scopes) should be included as it is of great importance for users (cf. the literature e.g. Martin et al., J Urol 2017, Al-Balushi et al. Int UrolNephrol 2019).
Answer: Thank you for your comment. We cited relevant researches, including those you mentioned, and added it to the discussion.
“Some studies have calculated the break-even point between disposable flexible ureteroscopes and reusable flexible ureteroscopes. [38,39]. In these studies, disposable flexible ureteroscopes are more cost-effective than reusable flexible ureteroscopes when there are less than 22 to 99 cases of ureteroscopic surgeries, favoring the use of disposable flexible ureteroscopes in smaller centers.” [Page 11]

Reviewer 2 Report (Previous Reviewer 2)
I congratulate you for coming up with this article.
Overall opinion - this article is well-written, concise and clear.
It is scientifically sound and contains sufficient interest and originality to merit publication. The article can be printed with the current conditions.
Author Response
Reviewer 2
I congratulate you for coming up with this article.
Overall opinion - this article is well-written, concise and clear.
It is scientifically sound and contains sufficient interest and originality to merit publication. The article can be printed with the current conditions.
Answer: Thank you for your comment.

This manuscript is a resubmission of an earlier submission. The following is a list of the peer review reports and author responses from that submission.
Round 1
Reviewer 1 Report
Dear Author (s)
1. It should separate the analysis based on study design (RCT, non-RCT, ...).
2. There are grammatical errors.
3. The search is not systematic. Because relevant studies such as meta-analyses related to the topic have not been cited. "Comparison of single-use and reusable flexible ureteroscope for renal stone management: a pooled analysis of 772 patients". "Reusable flexible ureterorenoscopes are more cost-effective than single-use scopes: results of a systematic review from PETRA Uro-group.". etc.
Author Response
Reviewer 1
Comment 1-1: It should separate the analysis based on study design (RCT, non-RCT, ...).
Answer 1-1: We agree with your comment and added a new forest plot using RCT and non-RCT in Figure 3.
“In the meta-analysis of SFR in each study design, heterogeneity was high and random-effects model was applied (Figure 3).” [Page 4]
“In a subgroup analysis of SFR according to study design, there was no difference of SFR between each study design of RCT, prospective study, and retrospective study (P = 0.14; OR = 1.36; 95% CI, 0.90 to 2.04; Figure 3). Also, there was no subgroup difference (P = 0.90).” [Page 8]
Comment 1-2: There are grammatical errors.
Answer 1-2: Before revision submission, our manuscript was already proofread by a native English speaker.
Comment 1-3: The search is not systematic. Because relevant studies such as meta-analyses related to the topic have not been cited. "Comparison of single-use and reusable flexible ureteroscope for renal stone management: a pooled analysis of 772 patients". "Reusable flexible ureterorenoscopes are more cost-effective than single-use scopes: results of a systematic review from PETRA Uro-group.". etc.
Answer 1-3: We cited relevant research, including those you mentioned, in the discussion.
“There was a previous study comparing disposable and reusable fURS using me-ta-analysis [49]. Although many studies on fURS have been conducted since that study, only a few studies were included in the current analysis, as the results of some studies were not relevant to our study [50]. The studies not including SFR, operation time, and length of hospital stay did not satisfy the inclusion criteria and were excluded. Among the studies not included, there were studies that dealt with important topics not covered in this study. First, the economic aspects of disposable and reusable fURS are important in that they influence the choice between disposable and reusable fURS. A previous study analyzed the cost per procedure of reusable flexible ureteroscope, and concluded that re-usable flexible ureteroscope is more cost-effective than disposable flexible ureteroscope [51]. Second, the importance of environmental aspects has been emphasized. A previous study examined the equivalent mass of carbon dioxide emitted each time a disposable or reusable flexible ureteroscope is used, and found that the environmental impact was comparable between disposable and reusable flexible ureteroscopes [52].” [Page 12]

Reviewer 2 Report
I congratulate you for coming up with this article. The subject of the work is of interest. However, there are some points that need consideration:
1) There are a few grammatical errors which make some sentences difficult to read so I would suggest another round of editing.
2) The title in not clear.
3) Please provide the keywords based on MeSH terms. (http://www.nlm.nih.gov/mesh/MBrowser.html)
4) The keywords should be sorted alphabetically.
Introduction- 5) Authors did not comment on the gap of knowledge the current research will try to fill.
6) Authors did not report related researches and what they were missing.
Result- 7) As a general and unwritten rule, for every 1,000 to 1,500 words, a table or image is considered a standard article. Due to the fact that some data is expressed in the text, it is recommended to reduce the number of tables and images.
Author Response
Reviewer 2
I congratulate you for coming up with this article. The subject of the work is of interest. However, there are some points that need consideration:
Comment 2-1: There are a few grammatical errors which make some sentences difficult to read so I would suggest another round of editing.
Answer 2-1: Before revision submission, our manuscript was already proofread by a native English speaker.
Comment 2-2: The title in not clear.
Answer 2-2: We changed the title as follows: “Comparison of surgical outcomes between single-use and re-usable flexible ureteroscope for renal stone management: a systematic review and meta-analysis.” [Page 1]
Comment 2-3: Please provide the keywords based on MeSH terms. (http://www.nlm.nih.gov/mesh/MBrowser.html). The keywords should be sorted alphabetically.
Answer 2-3: We provided MeSH term keywords and sorted them alphabetically according to your comment.
“Keywords: disposable equipment; kidney calculi; meta-analysis; systematic review; ureteroscopes” [Page 1]
Introduction
Comment 2-4: Authors did not comment on the gap of knowledge the current research will try to fill.
Answer 2-4: In this study, we tried to present updated results by adding recent studies. The evidence has been solidified, and we have obtained additional sub-analytical data on the fiber and digital reusable ureteroscopes. As you commented, the gap of knowledge that the current research will try to fill has been outlined in the discussion.
“This study will affirm the reliability of disposable flexible ureteroscopic stone surgeries, with the addition of updated evidence through meta-analysis of recent studies.” [Page 2]
Comment 2-5: Authors did not report related researches and what they were missing.
Answer 2-5: In some papers, the PICO inclusion criteria were not satisfied because there were no stone-free rate, operation time, and length hospital stay, which we wanted to analyze. Therefore, some papers were excluded, as described in the PRISMA study selection flow chart. The content of the missing studies you mentioned has been emphasized once again in the discussion.
“There was a previous study comparing disposable and reusable fURS using me-ta-analysis [49]. Although many studies on fURS have been conducted since that study, only a few studies were included in the current analysis, as the results of some studies were not relevant to our study [50]. The studies not including SFR, operation time, and length of hospital stay did not satisfy the inclusion criteria and were excluded. Among the studies not included, there were studies that dealt with important topics not covered in this study. First, the economic aspects of disposable and reusable fURS are important in that they influence the choice between disposable and reusable fURS. A previous study analyzed the cost per procedure of reusable flexible ureteroscope, and concluded that re-usable flexible ureteroscope is more cost-effective than disposable flexible ureteroscope [51]. Second, the importance of environmental aspects has been emphasized. A previous study examined the equivalent mass of carbon dioxide emitted each time a disposable or reusable flexible ureteroscope is used, and found that the environmental impact was comparable between disposable and reusable flexible ureteroscopes [52].” [Page 12]
Result
Comment 2-6: As a general and unwritten rule, for every 1,000 to 1,500 words, a table or image is considered a standard article. Due to the fact that some data is expressed in the text, it is recommended to reduce the number of tables and images
Answer 2-6: Some of the figures have been changed to supplementary files according to your comment.

Reviewer 3 Report
The authors have submitted a manuscript on a meta-analysis of studies comparing single-use and reusable flexible ureteroscopes for kidney and ureteral stone treatment. The number of studies that could be included was quite low (n=11). Only n=3 were randomized controlled trials.
The authors concluded that there were no differences in the SFR, complication, operation time, 325 and hospital stay between reusable and disposable fURS.
Two important aspects of using disposable instruments were not studied at all: 1. The economy, 2. The ecological footprint. Today, both issues play an increasing role which should not be neglected. There are several studies in the literature dealing with the economic aspects. For the ecological aspects, there are only few studies. Nevertheless, they should be discussed.
Author Response
Reviewer 3
Comment 3-1: The authors have submitted a manuscript on a meta-analysis of studies comparing single-use and reusable flexible ureteroscopes for kidney and ureteral stone treatment. The number of studies that could be included was quite low (n=11). Only n=3 were randomized controlled trials.
Answer 3-1: We completely agree with your comment. Quality assessment was done in each study design of RCT and non-RCT using Cochrane risk of bias and methodological index for non-randomized studies. Subgroup analysis was done according to the study design of RCT and non-RCT. Additionally, we performed subgroup analysis for each study design, RCT and non-RCT.
“Cochrane risk of bias (ROB) tool was used for RCTs, and the methodological index for nonrandomized studies (MINORS) was used for nonrandomized studies.” [Page 3]
“In the meta-analysis of SFR in each study design, heterogeneity was high and random-effects model was applied (Figure 3).” [Page 4]
“In a subgroup analysis of SFR according to study design, there was no difference of SFR between each study design of RCT, prospective study, and retrospective study (P = 0.14; OR = 1.36; 95% CI, 0.90 to 2.04; Figure 3). Also, there was no subgroup difference (P = 0.90).” [Page 8]
Comment 3-2: The authors concluded that there were no differences in the SFR, complication, operation time, 325 and hospital stay between reusable and disposable fURS. Two important aspects of using disposable instruments were not studied at all: 1. The economy, 2. The ecological footprint. Today, both issues play an increasing role which should not be neglected. There are several studies in the literature dealing with the economic aspects. For the ecological aspects, there are only few studies. Nevertheless, they should be discussed.
Answer 3-2: As you suggested, we should consider the economic and ecological aspects for disposable medical instruments and materials. Therefore, in the discussion, research on economic and ecological aspects have been described.
“There was a previous study comparing disposable and reusable fURS using me-ta-analysis [49]. Although many studies on fURS have been conducted since that study, only a few studies were included in the current analysis, as the results of some studies were not relevant to our study [50]. The studies not including SFR, operation time, and length of hospital stay did not satisfy the inclusion criteria and were excluded. Among the studies not included, there were studies that dealt with important topics not covered in this study. First, the economic aspects of disposable and reusable fURS are important in that they influence the choice between disposable and reusable fURS. A previous study analyzed the cost per procedure of reusable flexible ureteroscope, and concluded that re-usable flexible ureteroscope is more cost-effective than disposable flexible ureteroscope [51]. Second, the importance of environmental aspects has been emphasized. A previous study examined the equivalent mass of carbon dioxide emitted each time a disposable or reusable flexible ureteroscope is used, and found that the environmental impact was comparable between disposable and reusable flexible ureteroscopes [52].” [Page 12]
